# Evaluation of the effectiveness of combined staged surgical treatment in patients with keratoconus

**Polad M. Maharramov****\*, Fidan A. Aghayeva**

National Centre of Ophthalmology named after academician Zarifa Aliyeva, Baku, Azerbaijan

\* maharramov@mail.ru

**Data Availability Statement:** All relevant data are within the paper.

**Funding:** The authors received no specific funding for this work.

## Abstract

### Purpose

This study performs comparative assessment of the results of different types of two-stage surgical treatment in patients with keratoconus, including combination of corneal collagen cross-linking with intrastromal corneal ring segments followed by topography-guided photo-refractive keratectomy.

### Materials and methods

Prospective review of 101 patients (101 eyes) with keratoconus was performed. Patients underwent corneal collagen cross-linking (32 patients), intrastromal corneal ring segments (48 patients), and a combination of these two procedures (21 patients). Transepithelial topography-guided photorefractive keratectomy was performed as the second stage of treatment in all patients with obtained stable refractive results at 8 months after first stage. Main outcome measures were visual acuity (uncorrected distance and corrected distance) and corneal topographic indices.

### Results

Comparison of the studied parameters after first stage surgical treatment between non-combined CXL and combined groups demonstrated a statistically significant difference for uncorrected distance visual acuity, corrected distance visual acuity, and cylindrical refraction values ($p < 0.05$). We observed significant improvement of visual acuity and key corneal topographic indices after topography-guided photorefractive keratectomy in all study groups ($p < 0.05$). In 50 (49.5%) patients customized excimer laser ablation gave the possibility of full spherical and cylindrical corrections. Ten eyes (10%) had delayed epithelial healing, no corneal stromal opacities developed.

### Conclusions

This study shows that combined two-stage surgical treatment of keratoconus, consisting of intrastromal corneal ring segment implantation with corneal collagen cross-linking followed by topography-guided photorefractive keratectomy, is clinically more effective to prevent

**Competing interests:** The authors have declared that no competing interests exist.

keratectasia progression and increase visual acuity than the use of non-combined two-stage techniques.

## Introduction

Keratoconus is a progressive bilateral pathology of cornea, which is manifested by its thinning and protrusion that lead to astigmatism and deterioration of vision [1]. Different treatment options, such as spectacles and contact lenses [2], intrastromal corneal ring segments (ICRS) [3–7], keratoplasty (penetrating and deep anterior lamellar), corneal collagen cross-linking (CXL) [8–11] with topography-guided customized excimer laser ablation treatment—photorefractive keratectomy (PRK) [12–21], and combinations of the above-mentioned methods [16,21–23] with the application of novel ray-tracing excimer laser customization [24] could be used in patients with keratoconus, depending on its severity and grade. All treatment methods are reported to be effective and safe. This conclusion is based on the comparative analysis of such pre- and postoperative parameters, as visual acuity, refraction and corneal topographic indices. Several reports about the combination of CXL, ICRS, and topography-guided PRK (TG-PRK) were published [25]. The authors considered several combinations of different treatment methods, including phototherapeutic keratectomy (PTK) and implantation of phakic intraocular lenses [16,21]. It is supposed that corneal surgery in keratoconus should both provide prevention of the progression of pathology and also an improvement in vision [26]. An analytical review of literature indicates that the use of only CXL for the treatment of keratoconus has almost no effect on functional parameters, while effectively stops the disease progression, due to the cross-linking of corneal collagen with riboflavin and an increase in corneal biomechanics and strength [13]. The combination of CXL with PRK has a double effect: an increase in corneal stability and a significant improvement in visual acuity [12–21], but reported data is still controversial. Egyptian scientists presented the effectiveness of combination of CXL with PRK in correction of the refractive status in seventy-nine eyes of 46 patients with 18 months of follow-up. However, longer-term postoperative complications and a high rate of postoperative disease progression made confirmation of the safety and stability of the procedure challenging [12]. Interesting data about the clinical efficacy of the combination of ICRS with CXL in the treatment of keratoconus was presented by different researchers [4–7,23]. A more extended and detailed review of the protocols for the combined use of CXL (CXL plus) with different other treatment modalities demonstrates that further studies are required to determine the most effective treatment strategy [23]. Therefore, we decided to compare the results of different types of two-stage surgical treatment in patients with keratoconus, including combination of CXL with ICRS followed by TG-PRK.

## Materials and methods

We performed a prospective clinical study including 101 patients (101 eyes) with keratoconus who underwent two-stage surgical management at the National Centre of Ophthalmology named after academician Zarifa Aliyeva, Baku, Azerbaijan. Mean age of the patients was 20.1 ±8.4 years, 53 were males and 48 females. The occupations of treated patients included office workers (42 patients), oil workers (20 patients), doctors (15 patients), and other (24 patients). The study adhered to the tenets of the Declaration of Helsinki and was approved by the Ethics Committee of the National Centre of Ophthalmology named after academician Zarifa Aliyeva, Baku, Azerbaijan.

As the first stage of treatment, 32 patients (32 eyes) underwent only CXL (CXL group), 48 patients (48 eyes) underwent only ICRS (ICRS group) and 21 patients with keratoconus stage II–III in accordance with Amsler-Krumeich classification, underwent a combination of CXL and ICRS (first ICRS, after 24 hours CXL, ICRS+CXL group) from January 2017 to January 2018. Decision to select and include patients into one of the abovementioned three groups was made on a clinical basis (keratoconus stage and topographic parameters) and informed written consent was obtained in all patients. Transepithelial TG-PRK was performed as the second stage of treatment in all patients with obtained stable refractive results at 8 months after first stage.

Surgical technique was standard. KeraRing ICRS (Mediphacos, Belo Horizonte, Brazil) was implanted into corneal intrastromal tunnels with the inside and outside diameters 4.4 mm and 5.6 mm, respectively, created by means of the 200 kHz femtosecond laser FS200 (Alcon Wave-Light®, USA). The ring was implanted in the thinnest part of the cornea at a depth of 90 μm from the endothelium.

To perform CXL under topical anaesthesia, corneal deepithelialization was carried out with the use of a special scraper within the boundaries of the treatment zone of 7 mm, marked with a special trephine. Then 0.1% riboflavin solution (Riboflavin Medio Cross) was applied on the cornea every 5 minutes for 30 minutes to increase the absorption of ultraviolet-A light (UVA) radiation and exposure to UVA itself with a wavelength of 365 nm, radiation energy of 3.0 mW/cm$^2$ and light spot diameter of 7.0 mm (UVX-1000, IROCAG, Switzerland) in six steps of 5 minutes each was performed. A soft bandage contact lens was applied at the end of surgery. During transepithelial TG-PRK 50 μm thick corneal epithelium was removed by PTK under topical anaesthesia, then ablation of the stroma was performed by PRK. The planned maximal topographic stromal ablation depth did not exceed 50 μm after epithelium removal.

All patients were examined before surgery and at 1, 3, 6, 8 months after every surgical procedure during first and the second stage treatment (mean follow-up period was 16.2±1.8 months; minimal follow-up of patients in all groups was 14 months). The ophthalmic examination included measurement of uncorrected distance and corrected distance visual acuity (UDVA and CDVA), autorefractometry (TOMEY RC-5000, Japan), noncontact tonometry (TOMEY FT-1000, Japan), biomicroscopy (Tomey TSL 5000, Tomey, Japan), Pentacam corneal tomography (Oculus Wetzlar, Germany), Wavelight Oculyzer (ALCON, USA), Wave-Light® Topolyzer® VARIO (ALCON, USA), anterior segment OCT (Cirrus HD-OCT 5000, Zeiss, Germany), ultrasound pachymetry, and funduscopy.

### Statistical analysis:

Statistical analysis was conducted using analysis methods of quantitative and qualitative characteristics (descriptive statistics, one-factor and two-factor variance analysis, two sample t-test with different variances) with the use of an appropriate software package Microsoft Excel-2010 [27]. Continuous data are presented as mean ± standard deviation and chi-square test was used to compare pre- and postoperative parameters.

### Results

The preoperative corneal thickness was at least 400 μm at the thinnest corneal location in all patients. Data on visual acuity and corneal topography before surgery and 6 months after CXL, ICRS and ICRS with CXL are shown in Table 1. There were no statistically significant differences in preoperative visual acuity and corneal indices between groups. Presented data indicate that visual acuity and key corneal topographic indices changed noticeably at 6 months after all three types of first stage surgical treatment, thus confirming the achievement of the

**Table 1. Clinical parameters before and 6 months after CXL, ICRS and ICRS+CXL.**

| Mean indices | CXL | | ICRS | | ICRS+CXL | |
|---|---|---|---|---|---|---|
| | before | after | before | after | before | after |
| UDVA, decimal | 0.2±0.03 | 0.2±0.04* | 0.1±0.03 | 0.5±0.03* | 0.1±0.03 | 0.5±0.04*^ |
| CDVA, decimal | 0.4±0.04 | 0.4±0.06* | 0.4±0.03 | 0.8±0.03* | 0.3±0.04 | 0.9±0.03*^ |
| Cylindrical refraction | -5.2±0.1 | -4.8±0.1 | -5.4±0.1 | -2.1±0.1 | -6.2±0.1 | -1.5±0.1^ |
| Spherical refraction | -2.1±0.1 | -1.9±0.1 | -6±0.1 | -1.9±0.1 | -7.1±0.1 | -1.9±0.1 |
| Spherical equivalent, D | -5.6±0.1 | 5.1±0.1* | -5.5±0.1 | 5.2±0.1* | -5.6±0.1 | 5.1±0.1* |
| Keratometry of the anterior surface, a steep axis, D | 48.9±0.9 | 47.1±0.1* | 50.8±0.1 | 44.3±0.1* | 50.8±0.1 | 43.1±0.1 |
| Keratometry of the posterior surface, a steep axis, D | -7.7±0.1 | -7.8±0.1 | -7.7±0.1 | -7.8±0.1 | -7.7±0.1 | 7.8±0.1 |
| Thickness of cornea, apex, μm | 456±4.1 | 448±3.8 | 458±4.5 | 467±4.2 | 459±4.1 | 446±4 |
| Volume of cornea, mm$^3$ | 57±0.2 | 56.1±0.3* | 56.9±0.3 | 55.8±0.2* | 56.8±0.3 | 55.4±0.2* |
| Index of asphericity (Q) | -0.9±0.1 | -0.9±0.1 | -0.8±0.1 | -0.9±0.04 | -0.9±0.04 | -0.9±0.04 |
| Index of progression | 2.3±0.1 | 2.5±0.1* | 2.3±0.1 | 2.5±0.1* | 2.3±0.1 | 2.5±0.1* |
| Index of surface variance (ISV) | 95.5±5.6 | 75.4±2.8* | 99±3.1 | 76.5±3.3* | 98±2.9 | 74.5±3.1* |
| Index of vertical asymmetry (IVA) | 1.1±0.1 | 0.8±0.1* | 1±0.1 | 0.8±0.1* | 1±0.1 | 0.7±0.1* |
| Keratoconus index (KI) | 1.3±0.01 | 1.2±0.01* | 1.3±0.02 | 1.2±0.01* | 1.2±0.01 | 1.2±0.02* |
| Index of height asymmetry (IHA) | 26.2±1.3 | 23.8±1.2* | 26.4±1.5 | 24±1.2* | 26.5±1.3 | 22.9±1.2* |
| Surface regularity index (SRI) | 1.2±0.1 | 0.9±0.1* | 1.2±0.1 | 1±0.1* | 1.3±0.1 | 0.9±0.1* |
| Surface asymmetry index (SAI) | 2.9±0.1 | 2.5±0.1* | 3±0.1 | 2.6±0.1* | 3±0.1 | 2.7±0.1* |

CXL = corneal collagen cross-linking; ICRS = intrastromal corneal ring segments;

UDVA = uncorrected distance visual acuity; CDVA = corrected distance visual acuity.

*P < .05 (comparison with preoperative data in all groups); ^P$_1$ < .05 (comparison of postoperative data between ICRS+CXL and CXL groups)

expected effect (p<0.05). The clinical results after combination of CXL with ICRS are comparatively better than in CXL non-combined group; the differences for UDVA, CDVA, and cylindrical refraction values (p<0.05) between these groups were statistically significant. Clinical parameters before TG-PRK (8 months after first stage) and 6 months after TG-PRK are presented in Table 2.

These data confirm statistically significant improvement of visual acuity and key topographic indices of cornea after TG-PRK in all studied groups (p<0.05) (Figs 1–3). Several parameters, such as UDVA, cylindrical refraction, spherical equivalent, and IVA, were better in patients underwent previous combination of CXL and ICRS. In 50 (49.5%) patients TG-PRK gave the possibility of full spherical and cylindrical corrections. No intraoperative complications were observed in all groups of patients. Ten eyes (10%) had delayed epithelial healing, no corneal stromal opacities developed.

## Discussion

All of the modern surgical methods for the treatment of keratoconus have been developed considering its pathogenesis. Crosslinking is a photopolymerization of stromal collagen fibers of the cornea, caused by the combined effect of a photosensitizing substance (riboflavin or vitamin B2) and UVA exposure. The results of studies on the use of CXL show that the procedure is highly effective with a significant improvement of keratometric indices and a detectable increase of visual acuity at 12 months after surgery [8–11]. Our study demonstrates statistically significant increase in UDVA and CDVA, and a reduction of steep axis keratometry of the

**Table 2. Clinical parameters before TG-PRK (8 months after first stage) and 6 months after TG-PRK in patients underwent CXL, ICRS or combined treatment.**

| Mean indices | CXL+TG-PRK | | ICRS+TG-PRK | | ICRS+CXL+TG-PRK | |
|---|---|---|---|---|---|---|
| | before | after | before | after | before | after |
| UDVA, decimal | 0.2±0.04 | 0.6±0.03* | 0.5±0,03 | 0.6±0,03* | 0.5±0.04 | 0.7±0.04* |
| CDVA, decimal | 0.4±0.06 | 1±0.04* | 0.8±0,03 | 0.9±0,04* | 0.9±0.03 | 1±0.03* |
| Cylindrical refraction | -4.8±0.1 | -0.9±0.1* | -2.1±0,1 | -1.1±0,1 | -1.5±0.1 | -0.8±0.1* |
| Spherical refraction | -1.9±0.1 | -0.7±0.1* | -1.9±0,1 | -0.6±0,1 | -1.9±0.1 | -0.5±0.1* |
| Spherical equivalent, D | 5.1±0.1 | 4.9±0.1 | 5.2±0,1 | 4.8±0,1* | 5.1±0.1 | 4.1±0.1* |
| Keratometry of the anterior surface, a steep axis, D | 47.1±0.1 | 46.1±0.1 | 44.3±0,1 | 44.1±0.1 | 43.1±0.1 | 41.1±0.1 |
| Keratometry of the posterior surface, a steep axis, D | -7.8±0.1 | -7.9±0.1 | -7.8±0.1 | -7.9±0.1 | -7.8±0.1 | -7.8±0.1 |
| Thickness of cornea, apex, μm | 448±3.8 | 380.1±3* | 467±4.2 | 376±3.1* | 446±4 | 370±3* |
| Volume of cornea, mm$^3$ | 56.1±0.3 | 56.0±0.2* | 55.8±0.2 | 55.1±0.2* | 55.4±0.2 | 54.8±0.1* |
| Index of asphericity (Q) | -0.9±0.1 | -0.9±0.1 | -0.9±0.04 | -0.9±0.04 | 0.9±0.04 | 0.9±0.05 |
| Index of progression | 2.5±0.1 | 2.6±0.1 | 2.5±0.1 | 2.6±0.1* | 2.5±0.1 | 2.7±0.1* |
| Index of surface variance (ISV) | 75.4±2.8 | 65.1±1.9* | 76.5±3.3 | 66.4±2.5* | 74.5±3.1 | 62±2.5* |
| Index of vertical asymmetry (IVA) | 0.8±0.1 | 0.7±0.1* | 0.8±0,01 | 0.7±0.04* | 0.7±0.1 | 0.5±0.1* |
| Keratoconus index (KI) | 1.2±0.01 | 1.1±0.01* | 1.2±0.01 | 1.1±0.02* | 1.2±0.02 | 1.1±0.01* |
| Index of height asymmetry (IHA) | 23.8±1.2 | 20.1±1.1* | 24±1.2 | 20.6±1.2* | 22.9±1.2 | 18.2±0.9* |
| Surface regularity index (SRI) | 0.9±0.1 | 0.7±0.1* | 1±0.1 | 0.8±0.1* | 0.9±0.1 | 0.6±0.1* |
| Surface asymmetry index (SAI) | 2.5±0.1 | 2±0.1* | 2.6±0.1 | 2.1±0.1* | 2.7±0.1 | 1.8±0.1* |

CXL = corneal collagen cross-linking; ICRS = intrastromal corneal ring segments;

TG-PRK = topography-guided photorefractive keratectomy; UDVA = uncorrected distance visual acuity;

CDVA = corrected distance visual acuity.

*P < .05 (comparison with preoperative data in all groups).

anterior surface. Furthermore, statistically significant changes of such keratometric indices as ICV, IVA, KI, SRI, SAI were revealed.

A huge number of published works are devoted to the investigation of the efficacy and safety of combined methods in the treatment of keratoconus [1,5,6,12,16,28]. Coshkunseven et al. reported an efficacy and safety of combined treatment, including posterior chamber toric phakic Visian ICL implantation after Keraring ICRS implantation followed by CXL in a three-step procedure in 14 keratoconic eyes with extreme myopia and astigmatism [16]. These works present mainly the results of combined treatment and there are just a few studies comparing the efficacy of various types of combined treatment [7,23]. Hosny et al. presented the comparative results of two variations of combined surgical treatment in keratoconus, including different CXL types: a combination of intra-tunnel CXL with ICRS implantation and a combination of epithelium-off CXL with ICRS in 20 eyes. Simultaneous intra-tunnel CXL with implantation of ICRS showed early visual rehabilitation due to the absence of epithelial defect [5]. Our observation gave possibility to compare efficacy of the independent application of CXL or ICRS with the use of their combination. We have found no statistically significant differences in clinical parameters between ICRS+CXL and ICRS groups, while a greater improvement of visual acuity and decrease in cylindrical refraction value in combined group compared with CXL group was revealed.

The treatment of keratoconus in mild and moderate stages with the use of combination of the CXL, PRK and PTK has been well documented. Kanellopoulos showed safety and long-term efficacy of the partial TG-PRK in conjunction with CXL (Athens Protocol) up to 3 years

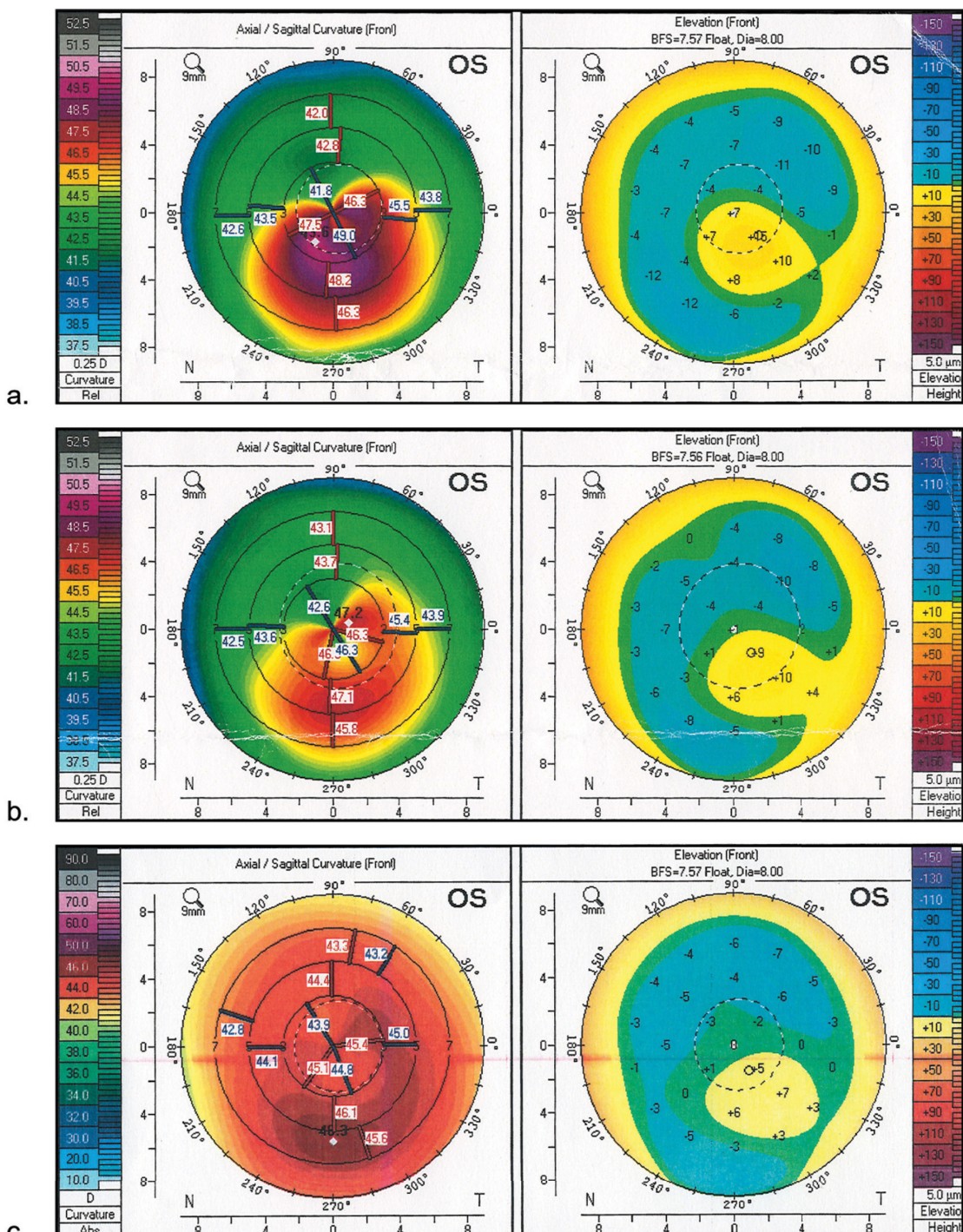

**Fig 1. Pentacam sagittal curvature (front) and elevation (front) maps in patient from CXL+TG-PRK group: a. initial keratometric maps; b. keratometric maps after CXL; c. keratometric maps after CXL+TG-PRK.** CXL = corneal collagen cross-linking; TG-PRK = topography-guided photorefractive keratectomy.

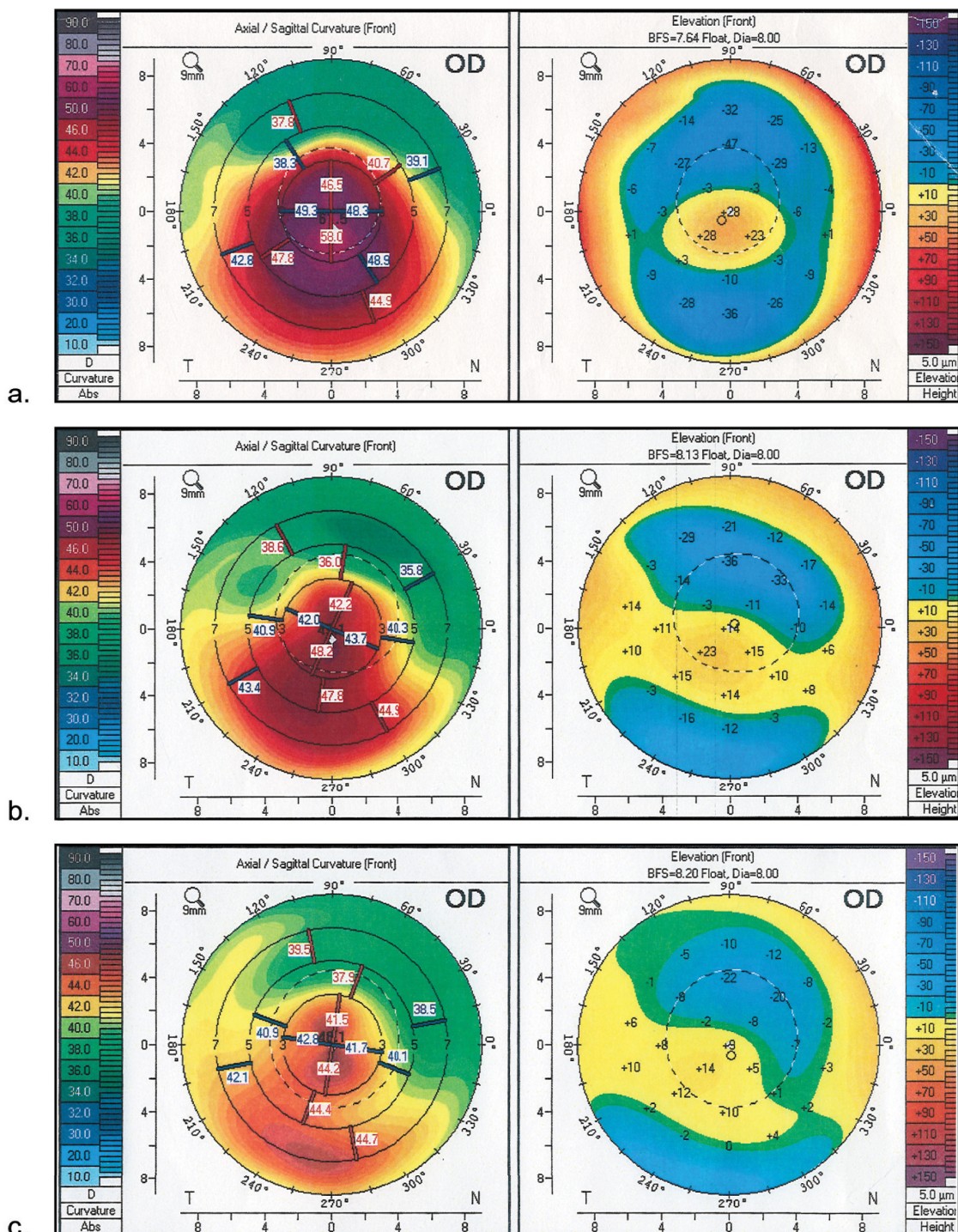

**Fig 2. Pentacam sagittal curvature (front) and elevation (front) maps in patient from ICRS+TG-PRK group: a. initial keratometric maps; b. keratometric maps after ICRS; c. keratometric maps after ICRS+TG-PRK.** ICRS = intrastromal corneal ring segments; TG-PRK = topography-guided photorefractive keratectomy.

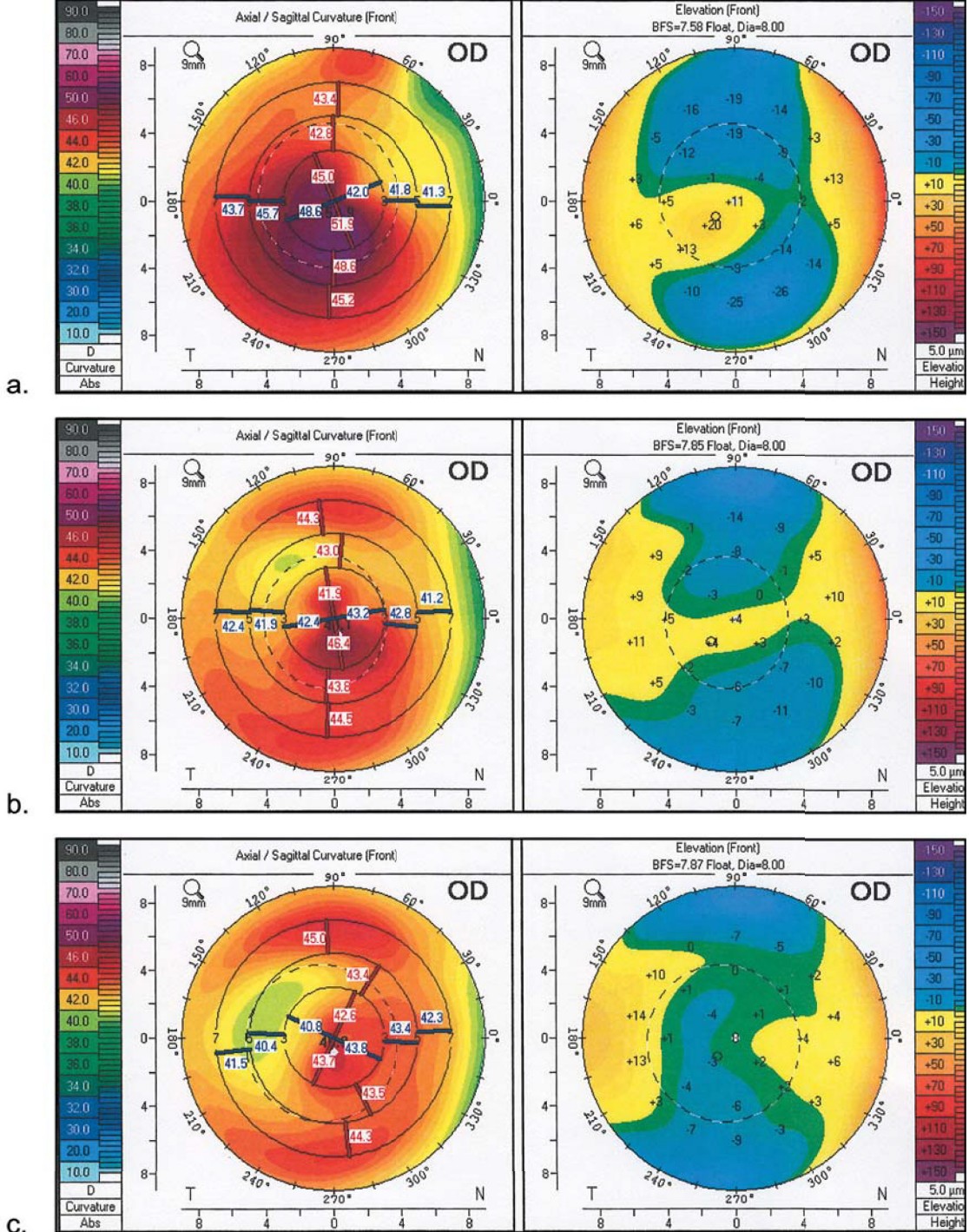

**Fig 3. Pentacam sagittal curvature (front) and elevation (front) maps in patient from ICRS+CXL+TG-PRK group: a. initial keratometric maps; b. keratometric maps after ICRS+CXL; c. keratometric maps after ICRS+CXL+TG-PRK.**
ICRS = intrastromal corneal ring segments; CXL = corneal collagen cross-linking; TG-PRK = topography-guided photorefractive keratectomy.

postoperatively in the treatment of two hundred thirty-one keratoconic cases [29] and at 10 years follow-up of one hundred thirty-four eyes with keratoconus [30]. Our study confirms that an application of TG-PRK in patients of all three groups provides further statistically significant improvement of visual acuity and key keratotopographic indices.

PRK-CXL combined refractive surgery treatment is considered to cause only several undesirable mild complications and identified as effective in slowing or arresting the progression of keratoconus [31]. Despite a plenty of publications on keratoconus treatment, there is still a debate, whether cross-linking and topographic ablation procedures should be performed simultaneously, sequentially and in what order [32,33]. In accordance with the results of Bardan et al., the use of CXL followed by sequential PRK provides a greater improvement in CDVA, SE, and refractive astigmatism [32]. It is well-known, that CXL causes thinning, due to an active increase of the degree of covalent bonding between and within collagen type I and proteoglycans and subsequent compaction of the collagen layers and cornea might not be suitable for further excimer laser ablation from a safety point of view [34]. Therefore, CXL performed after PRK is supposed to be safer. As the main purpose of our proposed treatment modality was to prevent keratectasia progression and stabilize corneal topography at first, we performed CXL as the first stage treatment in our keratoconic patients. TG-PRK was applied as an attempt to correct spherical and cylindrical errors in keratoconus only at already suitable stabilized cornea at 8 months after first stage. This approach gave the possibility of full spherical and cylindrical correction in almost half of our patients, better mean UDVA and CDVA values were achieved in ICRS+CXL+TG-PRK and CXL+TG-PRK/ ICRS+CXL+TG-PRK groups, respectively. One of the explanations could be that the majority of our patients were diagnosed with the II-III stage of keratoconus in accordance with Amsler-Krumeich classification, and preoperative corneal thickness was at least 400 μm at the thinnest corneal location in all patients.

One of the strengths of our study is that we compared not only the results of a combined keratoconus treatment method (CXL and ICRS) with non-combined options [1,5,7,12], but also different types of two-stage surgical treatment of keratoconus with the use of transepithelial TG-PRK. CXL and ICRS, as well as a combined version (CXL+ICRS) followed by TG-PRK are more clinically effective than methods without use of TG-PRK. By taking into account the confirmed greater efficacy of combined treatment CXL+ICRS, its staged use with TG-PRK is supposed to be the most optimal option in the treatment of keratoconus: CXL+ICRS as the first stage and TG-PRK as the second stage. These study results are considered to be reliable to prove the efficacy and safety of proposed keratoconus treatment strategy in terms of treated patients' quantity and number of analysed clinical parameters.

Thus, our data demonstrate that different options of two-stage treatment modality are effective and safe to stabilize ectasia and improve corneal regularity in eyes with mild to moderate keratoconus, so can improve quality of life in these patients. Combined two-stage (with 8-month interval) surgical treatment in patients with keratoconus, consisting of ICRS implantation with CXL followed by TG-PRK, is more effective to prevent keratectasia progression and increase visual acuity than the use of non-combined two-stage methods.

## Author Contributions

**Conceptualization:** Polad M. Maharramov.

**Data curation:** Polad M. Maharramov, Fidan A. Aghayeva.

**Formal analysis:** Polad M. Maharramov, Fidan A. Aghayeva.

**Investigation:** Polad M. Maharramov, Fidan A. Aghayeva.

**Methodology:** Polad M. Maharramov, Fidan A. Aghayeva.

**Project administration:** Polad M. Maharramov.

**Supervision:** Polad M. Maharramov.

**Validation:** Polad M. Maharramov.

**Visualization:** Polad M. Maharramov, Fidan A. Aghayeva.

**Writing – original draft:** Polad M. Maharramov, Fidan A. Aghayeva.

**Writing – review & editing:** Fidan A. Aghayeva.

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
