## [Decision Letter · Decision Letter 0]

1 Dec 2021

PONE-D-21-31417Evaluation of the effectiveness of combined staged surgical treatment in patients with keratoconusPLOS ONE

Dear Dr. %Maharramov%,

Thank you for submitting your manuscript to PLOS ONE. After careful consideration, we feel that it has merit but does not fully meet PLOS ONE’s publication criteria as it currently stands. Therefore, we invite you to submit a revised version of the manuscript that addresses the points raised during the review process.

We look forward to receiving your revised manuscript.

Kind regards,

Rajiv R. Mohan, Ph.D.

Academic Editor

PLOS ONE

Journal Requirements:

Additional Editor Comments:

Dear authors,

The reviewers raised significant concerns and AE agrees to them. I would like you to address these issues before reaching to conclusion for publication. Please pay careful attention to all concerns raised by two reviewers. It is likely that your revised manuscript will be returned to at least one more referee. Sometimes, an expert who was not part of the initial review process will also be invited to comment on the revision. Criticisms that were not mentioned during the initial review may arise at a future stage of the peer review process. Please pay careful attention to points raised by two reviewers.

Reviewers' comments:

Reviewer's Responses to Questions

**Comments to the Author**

1. Is the manuscript technically sound, and do the data support the conclusions?

Reviewer #1: Partly

Reviewer #2: No

2. Has the statistical analysis been performed appropriately and rigorously? 

Reviewer #1: No

Reviewer #2: No

3. Have the authors made all data underlying the findings in their manuscript fully available?

Reviewer #1: Yes

Reviewer #2: No

4. Is the manuscript presented in an intelligible fashion and written in standard English?

Reviewer #1: Yes

Reviewer #2: No

5. Review Comments to the Author

Reviewer #1: Thank you for the submission about comparison of three types of keratoconus treatments and staged TG-PRK.

-> How were the patients divided into groups. Was it randomization? Was the selection and grouping done based on tomography parameters.

98 - please mention the method of removing the epithelium

100 - Please mention the UVA fluence and elaborate on the "6 steps of 5 minutes". Is that the routine practice at the authors' institute? Or was it done for the study patients only.

103-104 - Mention the PRK criteria - was there a cut off for the thinnest pachymetry? How much of the spherical and cylindrical correction was done - full or partial?

Table 1 - please mention that the visual acuity is in decimal form to avoid confusion with logMAR

Table 1 is quite busy - maybe split to vision/refraction and tomography parameters for better readability

Same suggestions for table 2.

What is the time points compared in Table 2? Because there is a fair amount of variation between the Table 1 post-therapy and Table 2 pre-therapy.

It appears that the data in second set of table 1 is 6 months post therapy and the first set of data in table 2 is 8 months post therapy (just before the PRK). However, for a difference in 2 months, some of the parameters have changed drastically. For example the cylinder in the CXL only group.

Please clarify the time points and verify the data.

The clinical picture of ICRS maybe omitted.

It would be nice to have comparative pentacam maps of the three study arms at pre-op, stage 1 and stage 2.

Please add references from the Kennelopolous group who proposed the Athens protocol and has long term outcomes.

There are only few studies where PRK is done after crosslinking. Usually the laser is done prior to the crosslinking. Two reasons - the ablation rate of crosslinked cornea may not be same as that of treatment naive cornea so the excimer laser outcomes maynot be predictable. Furthermore, crosslinking causes thinning due to compaction of the collagen layers. This might cause the cornea to be not suitable for excimer laser from a safety point of view.

This aspect needs to be mentioned a bit more in the discussion.

Reviewer #2: Author Comments:

1. Rewrite the manuscript, grammatically correct.

2. Add references to introduction.

3. Put lines in table I and table II.

4. Put pre operative topography in results.

5. Rewrite discussion with relevant references and proof reading of English grammar.

6. Total number of references are 11, please add relevant references with Vancouver manner.

6. PLOS authors have the option to publish the peer review history of their article (what does this mean?). If published, this will include your full peer review and any attached files.

Reviewer #1: No

Reviewer #2: **Yes: **Dr. Zaman Shah

---

## [Author Response · Author response to Decision Letter 0]

19 Jan 2022

Response to Reviewers

We would like to thank Editor and all the reviewers for their comments and recommendations.

Additional Editor Comments:

Dear authors,

The reviewers raised significant concerns and AE agrees to them. I would like you to address these issues before reaching to conclusion for publication. Please pay careful attention to all concerns raised by two reviewers. It is likely that your revised manuscript will be returned to at least one more referee. Sometimes, an expert who was not part of the initial review process will also be invited to comment on the revision. Criticisms that were not mentioned during the initial review may arise at a future stage of the peer review process. Please pay careful attention to points raised by two reviewers.

Our response to Editor:

Thank you very much for your attention and comment. Our manuscript has been revised in accordance with the PLOS ONE's style requirements. We have tried to address all issues and to pay careful attention to all concerns raised by two reviewers.

Reviewers' comments:

Reviewer's Responses to Questions

Comments to the Author

1. Is the manuscript technically sound, and do the data support the conclusions?

Reviewer #1: Partly

Reviewer #2: No

2. Has the statistical analysis been performed appropriately and rigorously? 

Reviewer #1: No

Reviewer #2: No

3. Have the authors made all data underlying the findings in their manuscript fully available?

Reviewer #1: Yes

Reviewer #2: No

4. Is the manuscript presented in an intelligible fashion and written in standard English?

Reviewer #1: Yes

Reviewer #2: No 

5. Review Comments to the Author and Responses to Them:

Reviewer #1: Thank you for the submission about comparison of three types of keratoconus treatments and staged TG-PRK.

Our responses to Reviewer #1:

-> How were the patients divided into groups. Was it randomization? Was the selection and grouping done based on tomography parameters.

We thank the reviewer for the valuable comments. All the patients were selected and divided into three groups based on a clinical basis (on the topographic parameters and keratoconus stage). This information has been added to the article (Material and Methods section, page 6, line 126-128). 

98 - please mention the method of removing the epithelium

Corneal deepithelialization was carried out with the use of a special scraper within the boundaries of the treatment zone of 7 mm, marked with a special trephine (Materials and Methods section, page 6, line 140-142).

100 - Please mention the UVA fluence and elaborate on the "6 steps of 5 minutes". Is that the routine practice at the authors' institute? Or was it done for the study patients only.

Thanks for this comment. The UVA fluence was mentioned and elaboration on the "6 steps of 5 minutes" was added to the article (Materials and Methods section, page 6, line 144-146). This is not the routine practice at our institute, but most of our surgeons use it for their keratoconic patients.

103-104 - Mention the PRK criteria - was there a cut off for the thinnest pachymetry? 

The planned maximal topographic stromal ablation depth did not exceed 50 �m after epithelium removal, as the preoperative corneal thickness was at least 400 µm at the thinnest corneal point in all patients and the minimum thickness of cornea after PRK should be 350 �m with the epithelium (Materials and Methods section, page 7, line 150-151). 

How much of the spherical and cylindrical correction was done - full or partial?

Full spherical and cylindrical corrections were achieved in almost 50% of our patients (Results section, page 10, line 187-188).

Table 1 - please mention that the visual acuity is in decimal form to avoid confusion with logMAR

Thanks for this notice. The type of visual acuity score (in decimal) has been mentioned in Table 1 and 2.

Table 1 is quite busy - maybe split to vision/refraction and tomography parameters for better readability

Same suggestions for table 2.

Thanks a lot. To have the clinical parameters altogether, we created only two tables. Additional horizontal lines have been put for better readability. In case, they look still busy, we could split them.

What is the time points compared in Table 2? Because there is a fair amount of variation between the Table 1 post-therapy and Table 2 pre-therapy.

It appears that the data in second set of table 1 is 6 months post therapy and the first set of data in table 2 is 8 months post therapy (just before the PRK). However, for a difference in 2 months, some of the parameters have changed drastically. For example the cylinder in the CXL only group.

Please clarify the time points and verify the data.

We totally agree with your critical comment. Several parameters were not correctly statistically analysed and written down and some of them shifted in Table 2. After appropriate and rigorous statistical analysis the presented data have been verified and changed in appropriate cells of Table 2. Time points for clinical parameters before (preoperative) and 6 months after CXL, ICRS and ICRS + CXL (6 months postoperative) were presented in the Title of Table 1. Transepithelial TG-PRK was performed at 8 months after first stage of the treatment (Material and Methods section, page 6, line 129-131). To further clarify the time points for clinical parameters before TG-PRK (8 months after first stage) and after TG-PRK (6 months after) in patients underwent CXL, ICRS or combined treatment, those were added in the Results section, page 6, line 134 and in the Title of Table 2.

The clinical picture of ICRS maybe omitted.

It would be nice to have comparative pentacam maps of the three study arms at pre-op, stage 1 and stage 2.

According to your recommendation the clinical picture of ICRS has been omitted and we have added three figures – one case in each of the three groups (CXL, ICRS and ICRS+CXL) to compare Pentacam maps of the three study arms at pre-op, stage 1 and stage 2. Corrected legends are indicated in Results section, page 10-11, line 191-207.

Please add references from the Kennelopolous group who proposed the Athens protocol and has long term outcomes.

Thanks for this valuable comment. We have mentioned long-term outcomes of the Athens Protocol proposed by Kanellopoulos et al. in the Discussion section (Discussion section, page 12, line 238-241) and added the corresponding references as well.

There are only few studies where PRK is done after crosslinking. Usually the laser is done prior to the crosslinking. Two reasons - the ablation rate of crosslinked cornea may not be same as that of treatment naive cornea so the excimer laser outcomes maynot be predictable. Furthermore, crosslinking causes thinning due to compaction of the collagen layers. This might cause the cornea to be not suitable for excimer laser from a safety point of view.

This aspect needs to be mentioned a bit more in the discussion.

Thanks for this notice. As it was mentioned from your side and is well known, crosslinking causes corneal thinning, due to compaction of the collagen layers and cornea might not be suitable for further excimer laser ablation from a safety point of view. Therefore, most surgeons prefer to perform CXL after PRK. The main purpose of proposed treatment modality in our study was to arrest keratectasia progression and stabilize corneal topography, therefore we performed CXL as first stage treatment in our keratoconic patients. We performed topography guided PRK only in suitable stabilized corneas. TG-PRK was applied as a method of spherical and cylindrical corrections at already stabilized cornea at 8 months after first stage. This approach gave the possibility of full spherical and cylindrical correction in almost half of our patients. This information has been included in the Discussion section of the manuscript (Discussion section, page 12-13, line 244-258).

Reviewer #2: Author Comments:

1. Rewrite the manuscript, grammatically correct.

We thank the reviewer for the valuable comments. The manuscript has been proofread, grammatically corrected, and some parts of it were totally rewritten.

2. Add references to introduction.

Thank you very much for this comment. Additional references have been added to Introduction section (Introduction section, page 4, line 75-80).

3. Put lines in table I and table II.

Thanks a lot. We have put the lines in both tables.

4. Put preoperative topography in results.

Preoperative and postoperative key topographic corneal parameters, such as steep keratometry of the anterior and posterior surfaces; surface regularity index (SRI); surface asymmetry index (SAI) and index of asphericity (Q) were shown in Table 1 and 2. We have also included preoperative and postoperative sagittal curvature and elevation maps for one case in each of the three groups (Figs 1-3). 

5. Rewrite discussion with relevant references and proof reading of English grammar.

Thanks for this comment. We have totally rewritten the Discussion section with proof reading of English grammar and added relevant references.

6. Total number of references are 11, please add relevant references with Vancouver manner.

We have increased the number of relevant references up to 34 with Vancouver manner, according to the submission rules of Plos One Journal. 6. PLOS authors have the option to publish the peer review history of their article (what does this mean?). If published, this will include your full peer review and any attached files.

Do you want your identity to be public for this peer review? For information about this choice, including consent withdrawal, please see our Privacy Policy.

Reviewer #1: No

Reviewer #2: Yes: Dr. Zaman Shah

---

## [Decision Letter · Decision Letter 1]

2 Feb 2022

Evaluation of the effectiveness of combined staged surgical treatment in patients with keratoconus

PONE-D-21-31417R1

Dear Dr. %Maharramov%,

We’re pleased to inform you that your manuscript has been judged scientifically suitable for publication and will be formally accepted for publication once it meets all outstanding technical requirements.

Kind regards,

Rajiv R. Mohan, Ph.D.

Academic Editor

PLOS ONE

Additional Editor Comments (optional):

Reviewers' comments:

Reviewer's Responses to Questions

**Comments to the Author**

1. If the authors have adequately addressed your comments raised in a previous round of review and you feel that this manuscript is now acceptable for publication, you may indicate that here to bypass the “Comments to the Author” section, enter your conflict of interest statement in the “Confidential to Editor” section, and submit your "Accept" recommendation.

Reviewer #1: All comments have been addressed

Reviewer #2: All comments have been addressed

2. Is the manuscript technically sound, and do the data support the conclusions?

Reviewer #1: Partly

Reviewer #2: Yes

3. Has the statistical analysis been performed appropriately and rigorously? 

Reviewer #1: No

Reviewer #2: Yes

4. Have the authors made all data underlying the findings in their manuscript fully available?

Reviewer #1: Yes

Reviewer #2: Yes

5. Is the manuscript presented in an intelligible fashion and written in standard English?

Reviewer #1: Yes

Reviewer #2: No

6. Review Comments to the Author

Reviewer #1: It would be quite helpful to have the manuscript revised by some English editing service, or have it looked at by a native English speaker. This will improve overall readability and grammar.

Title suggestion: Evaluation of the effectiveness of single and combined surgical treatments with staged photorefractive keratectomy in patients with keratoconus.

168: Since this is the first usage of IVA, please give the full form.

Some calculations appear to be a bit off. Please re-check and make corrections if necessary.

Table 1: Please calculate the Spherical Equivalent in ICRS and combined groups. Going by the mean spherical and cylindrical refractions given, the the C3R .

Table 1: UDVA in CXL before and after appear quite close, so how is the change significant?

Table 2: Please calculate the spherical equivalent in post-PRK in all groups. Should be less than 4.9 given the mean spherical and cylindrical values.

Reviewer #2: some minor correction to be made regarding grammar correction and other corrections. Add conclusion (heading) at the end of discussion.

7. PLOS authors have the option to publish the peer review history of their article (what does this mean?). If published, this will include your full peer review and any attached files.

Reviewer #1: No

Reviewer #2: **Yes: **Dr. Zaman Shah

---

## [Editor Report · Acceptance letter]

28 Feb 2022

PONE-D-21-31417R1 

Evaluation of the effectiveness of combined staged surgical treatment in patients with keratoconus 

Dear Dr. Maharramov:

I'm pleased to inform you that your manuscript has been deemed suitable for publication in PLOS ONE. Congratulations! Your manuscript is now with our production department. 

Kind regards, 

on behalf of

Dr. Rajiv R. Mohan 

Academic Editor

PLOS ONE